# The Effectiveness of an Alloplastic Epidermal Substitute in the Treatment of Burn Wounds in Children: A Comparative Clinical Study of Skin Substitutes and Silver and Paraffin Gauze Dressings

**DOI:** 10.3390/jcm13237238

**Published:** 2024-11-28

**Authors:** Aleksandra Barbachowska, Tomasz Korzeniowski, Agnieszka Surowiecka, Piotr Tomaka, Magdalena Bugaj-Tobiasz, Maciej Łączyk, Zofia Górecka, Anna Chrapusta, Jerzy Strużyna

**Affiliations:** 1East Center of Burns Treatment and Reconstructive Surgery, 21-010 Lęczna, Poland; aleksandrabarbachowska@gmail.com (A.B.); dr.surowiecka@gmail.com (A.S.); magdalena.bugaj@tlen.pl (M.B.-T.); laczyk.maciej@gmail.com (M.Ł.); zfgrecka@gmail.com (Z.G.); jerzy.struzyna@gmail.com (J.S.); 2Department of Plastic, Reconstructive Surgery and Burn Treatment, Medical University of Lublin, 20-093 Lublin, Poland; 3Department of Plastic, Reconstructive Surgery and Microsurgery, Medical University of Lublin, 20-093 Lublin, Poland; 4Department of Anesthesiology and Intensive Care, Independent Public District Hospital in Leczna, 21-010 Leczna, Poland; p.tomaka@szpital.leczna.pl; 5Malopolska Burn and Plastic Surgery Center, Ludwik Rydygier Memorial Hospital in Krakow, 31-820 Krakow, Poland; anna.chrapusta@gmail.com

**Keywords:** wound healing, burn, skin substitute

## Abstract

**Background**: Children make up a large percentage of those affected by burns worldwide, with most of them suffering from severe injuries that necessitate skilled medical attention. Despite medical progress, there is still no ideal dressing for the treatment of burn wounds in children. The aim of the study was to assess the impact of epidermal substitutes in the treatment of burn wounds in children. **Materials and Methods**: This retrospective study evaluates the use of three dressings in the treatment of pediatric burns at a major Polish burn center. A patient database was used to identify children who received treatment with silver dressings, paraffin dressings or epidermal substitutes from 2009 to 2023. A demographic analysis was performed to collect the following information: causes of burns, procedural details and patient outcomes. **Results**: There were 439 patients aged between 1 month and 18 years. For severe burns, the number of interventions was lowest among children with epidermal substitute application (*p* = 0.039). Paraffin gauze resulted in the greatest number of skin grafts, whereas alloplastic replacement produced the least amount of transplantation (*p* < 0.005) regardless of the severity of the burn. **Conclusions**: Epidermal substitutes offer a good dressing option for burn wounds to improve their treatment and reduce the need for skin graft coverage. In the future, extended comparative or randomized trials are needed to confirm our results.

## 1. Introduction

A World Health Organization (WHO) initiative called the Global Burn Registry (GBR) estimates that burn injuries cause over 180,000 fatalities annually [1]. Children make up a large percentage of those affected by burns worldwide, with most of them suffering from severe injuries that necessitate skilled medical attention. According to estimates, burns rank as the sixth most frequent non-fatal injury in children. Nonetheless, over the course of several decades, child mortality from burns has decreased, particularly in high-income nations. However, compared to that in high-income countries, the death rate of children is currently more than seven times greater in low- and middle-income countries [2]. Non-fatal burns frequently cause disfigurement, incapacity and extended hospital stays. The overall cost of hospitalization increases significantly as a result of the need for multimodal rehabilitation and specialized therapy. It is estimated that, in Poland, children under the age of 4 account for three-quarters of all burn cases; nevertheless, there are insufficient population-based reports that address burns among Polish children [3,4].

Children’s burn injuries differ from adult burns in that they exhibit distinct patterns. Toddlers and, in particular, infants have immature immune systems and have not been exposed to many virulent infections. As a result, they are more susceptible to infections and sepsis. Children have a larger body surface area-to-mass ratio, making them more susceptible to hypothermia. Their body surface area is distributed differently, making the head proportionally much larger compared to the limbs. Their skin is also different from that of adults. Because children have thinner skin, burns become deeper at the same temperature. In addition, their thinner skin has fewer appendages, which increases the likelihood of third-degree burns [5]. The unique pathological features of burn injuries in children combined with their unique physiological skin structure cause the burn to become more deeply converted into full-thickness wounds, which in turn cause hypertrophic scars to grow throughout the subsequent phases of treatment. Moreover, growth retardation may result from burn injuries that disturb metabolic balance. Problems with physiology and psychology have a significant effect on patients and their families [6,7,8].

The most frequent type of burn injury mechanism is usually a scald. Most of these burns are caused by hot liquid incidents that occur during domestic tasks [6,9]. Among teenagers, flame burns are typically the main reason for hospitalization. Burns resulting from explosions, direct contact with hot objects, hot oil or chemical exposure seem to be less typical [5,6].

The management of burns in children ranges from intensive emergency care and surgery to outpatient care. It is expected that 90% of children with burns will be hospitalized. Child mortality from burn injuries has significantly decreased as a result of major advancements in pediatric burn care over the past few decades, including early fluid resuscitation, balanced infection control and precise wound management [8,9,10,11].

Burn wound care includes debridement, preventing infection and providing a moist environment to promote wound healing. Currently, we can choose from a range of various dressings dedicated to the treatment of burn wounds. Most of them provide moisture and have antibacterial properties. In children, it is crucial that the dressing is comfortable to apply and remove, easy to use and has a long shelf life [12].

This has led to the development of many novel products for acute burn wounds, including hydrogels, hydrocolloids, alginates, silver dressings and biosynthetic skin substitutes [13]. The epidermal substitute, due to maintaining an optimal healing environment and plasticity that makes it easy to use, is an alternative for the treatment of burns, especially in children [14].

The aim of the study was to assess the impact of an alloplastic epidermal skin substitute used on the treatment of burn wounds among children in relation to length of hospitalization, the number of dressing changes and the need for skin transplantation.

## 2. Materials and Methods

### 2.1. Study Design

The documentation of 439 children treated for acute burns at the East Centre of Burns Treatment and Reconstructive Surgery in Leczna (Poland) in the years 2009–2023 was analyzed. The demographic data, etiology, area and degree of the burn, as well as the methods and effects of treatment, were analyzed.

### 2.2. Inclusion and Exclusion Criteria

The inclusion criteria were age between 1 month and 18 years and the presence of an acute burn up to 5 days after the incident. Individuals who did not meet the inclusion requirements, such as age over 18 years, old burns and children discharged from the hospital at the request of their parents, were excluded from the study.

### 2.3. Division of the Study Group According to the Severity of Burns

The group was split into three categories based on the severity of the burns: mild, moderate, and severe. The mild burn group included children with first-degree and second-degree burns covering less than 10% of the body, the moderate group included those with second-degree burns covering 10–20% of the body surface or third-degree burns below 10% and the severe group include those with second-degree burns covering more than 20% of the TBSA or third-degree burns covering more than 10% of the TBSA.

The assessment of burn depth was based on a clinical examination and an assessment of the visual and tactile characteristics of the burn wound (wound appearance, capillary refill and burn wound sensitivity). A division into 3 degrees of depth was used: superficial burn (first degree), patrial-thickness burn (second degree) and full-thickness burn (third degree).

### 2.4. Wound Management

Immediately after the injury, the wounds were protected with a hydrogel dressing by the medical rescue team or in the emergency department. Then, the wound was assessed for the depth and extent of the burn. A debridement procedure was performed no later than the first day after injury, under opiate or general anesthesia in the operating theater. This involved a thorough cleansing of the wound and the removal of blisters and keratin remnants.

### 2.5. Dressing Selection

The wounds were covered with one of three dressings, depending on the availability and preferences of the operating surgeon: paraffin tulle gras (Jelonet™, Smith & Nephew, Watford, UK), silver dressing (Aquacel^®^ Ag, ConvaTech Group, London, UK) or an epidermal substitute (Suprathel^®^, PolyMedics Innovations GmbH, Denkendorf, Germany). Gauze was used as a protective top dressing followed by bandage and elastic dressing mesh.

### 2.6. Intervention

The wounds were inspected every 2–3 days and the top layers of the dressing were replaced until epithelialization occurred. If spontaneous healing did not progress within 14 days, the wounds were covered with split-thickness skin grafting. The exception were 3rd degree wounds, where it was decided to immediately remove the necrotic tissue (enzymatic debridement or surgical necrectomy) and perform skin transplantation.

### 2.7. Characteristic of Dressings Used in the Study

Suprathel^®^ (PolyMedics Innovations GmbH, Denkendorf, Germany) was used as an alloplastic epidermal substitute in our study. This is a synthetic copolymer made from DL-lactide (>70%), trimethylene carbonate and e-caprolactone. The production process includes the polymerization of monomers in the melting procedure which are then dissolved in organic solvents. The resulting material undergoes a process of suitably modified phase inversion and a lyophilization technique. The result is a product in the form of a microporous membrane [15,16,17]. Thanks to its structure, it prevents the accumulation of secretions in the wound while providing a moist environment. Its great advantage is its high plasticity, which allows the dressing to be adjusted to the wound in various parts of the body [18].

Standard silver and paraffin dressings have also proven to be efficacious in advancing the wound healing process through a variety of mechanisms. The effectiveness of wound dressings can significantly impact patient outcomes, particularly in complex cases such as burns. 

Paraffin dressings provide a moist environment that promotes natural healing processes while preventing the dressing from sticking to the wound bed [19]. This feature is particularly beneficial in managing burns and skin grafts, where minimizing trauma during dressing changes is crucial [20]. Most are available in the form of tulles soaked in soft paraffin or chlorhexidine. Low-adherent dressing such as paraffin ones are usually cheap and widely available. In our study, Jelonet^®^ (Smith & Nephew, Watford, UK), a cotton leno fabric impregnated with white soft paraffin was used as the paraffin gauze dressing.

Silver dressings, infused with ionic or nanocrystalline silver, offer potent antimicrobial activity against a broad spectrum of pathogens, including antibiotic-resistant bacteria. This makes them invaluable in treating wounds at high risk of infection or those that are already infected [21]. Silver dressings not only help by reducing microbial load but also play a role in modulating inflammation and promoting a conducive environment for tissue regeneration [22,23]. The silver-containing dressing Aquacel^®^ Ag (ConvaTech Group, London, United Kingdom) was utilized in our study.

### 2.8. Ethical Statement

The Declaration of Helsinki’s principles were respected. The study received approval by the Institutional Ethics Committee of the Independent Public District Hospital in Leczna (ref. number: 02/WCLO/2023) approval date 20 March 2023.

### 2.9. Statistical Analysis

A comparative analysis was performed to investigate the association between the type of dressing and the duration of hospital stay. The Shapiro–Wilk test was used to verify the normality hypothesis. Given that the variables under analysis, like the number of hospital days and procedures, did not follow a normal distribution and showed a heterogeneity of variance, comparisons were made using the non-parametric Kruskal–Wallis test, which is the ANOVA test’s equivalent. Using Levene’s test, the hypothesis on the homogeneity of error variance was confirmed. The effect size was evaluated using the coefficient \(\hat{\varepsilon}_{ordinal}^2^\). Because of the large number of comparisons, Dunn’s post hoc test with Holm’s correction was employed when the dressing effect in the Kruskal–Wallis test reached statistical significance. The correction was applied in order to keep the type I error below the 0.05 threshold. The chi2 test was used to examine correlations when both the independent and dependent variables were measured on an ordinal or nominal scale. Cramer’s \(V\) coefficient was employed in this instance to evaluate the effect size. The statistical analysis was performed with Statistica^®^ 13.3 software (StatSoft, Cracow, Poland). Violin plots were superimposed on box-and-whisker plots for all comparative analyses. Using libraries that increased the functionality of the base package, statistical analyses and visualizations were carried out in the R statistical environment.

## 3. Results

The group’s features are displayed in Table 1.

The majority of burns affected the trunk (81%), followed by the upper extremity (37%) and head (34%).

In contrast to second degree burns (mean age 52), mosaic burns (mean age 50) and superficial burns (mean age 35), third degree burns were related to a higher age, mean 70 months, *p* < 0.001 (Figure 1). Furthermore, a simultaneous injury from inhalation and flames was more frequent with third-degree burns, *p* < 0.001.

For scalds and burns above 10% of the TBSA, the alloplastic epidermal substitute was most frequently applied. For flame injuries over 10% of the TBSA, paraffin gauze was chosen, *p* < 0.001.

The hospital stay (LOS) for superficial burns was the shortest (mean 8 days), while the longest LOS was seen in cases of third-degree burns (mean 20.56 days). When an epidermal substitute was utilized, the LOS was reduced in the subgroup of children with severe burns (*p* = 0.039). The use of nano-silver dressing as the initial treatment was found to result in shorter hospital stays across the whole study group (Figure 1).

The use of alloplastic epidermal substitutes was associated with more interventions for less extensive burns (<10% TBSA), *p* = 0.002, and fewer interventions for severe burns, *p* = 0.079.

Skin grafts were most frequently utilized for burns that were deeper than partial thickness and burns that were over 10% of the TBS (*p* < 0.001), most frequently as a result of flame injuries (*p* < 0.005). Additionally, as a result of flame damage, enzymatic debridement was carried out, *p* < 0.001. Paraffin gauze resulted in the greatest number of skin grafts, whereas alloplastic replacement produced the least amount of skin grafts, *p* < 0.005, regardless of the severity of the burn (Figure 2).

No adverse or allergic reaction to any of the dressings (Jelonet™, Aquacel^®^ Ag, Suprathel^®^) were recorded in the study group.

## 4. Discussion

In the treatment of pediatric burn patients, one of the most crucial steps is early and sustainable wound management. The choice of an appropriate local treatment depends primarily on the depth of the burn. The general management and healing of a burn wound were positively impacted not only by the use of common dressing types like paraffin- or silver-based ones, but also by the use of skin substitutes [24]. In deep dermal and full-thickness burns, enzymatic debridement, skin grafting and eschar excision are still essential procedures [25,26,27,28]. An alternative to traditional wound dressings might be an alloplastic epidermal substitute, which can be applied to superficial and deep partial-thickness burns [16].

When it comes to the group of adult patients, there are many studies comparing the effectiveness of various dressings in the treatment of burn wounds. Most of these studies focus on examining the effectiveness of dressings in terms of reducing pain and accelerating the epithelialization process [15,29,30]. It has also been established that an alloplastic skin substitute reduces the chance of infection by erecting a physical barrier against microbes and promotes wound healing by promoting angiogenesis and additional skin vasculogenesis. It is possible to prevent painful dressing changes because the alloplastic membrane fully biodegrades in 4–6 weeks [18,31]. German studies have also shown the usefulness of an epidermal substitute in treating partial-thickness burns in adults, but without any advantage over the epicyte epicite^hydro^, which consists of biotechnologically generated bacterial nanocellulose (BNC) synthesized by Komagataeibacter xylinus and 95% water. Both dressings were comfortable to use, flexible, provided an optimal healing environment and showed good esthetic and functional results without the need to change the dressing. Similarly, patients from our study group did not require changes to the epidermal substitute. Once applied, it allowed for spontaneous healing in partial-thickness burns. The wound was checked every few days, with only the secondary dressing being replaced, which avoided tearing the dressing material from the wound itself, reducing pain and thus increasing the comfort of the procedure [32].

Considering children’s burns, currently, there are no clear guidelines for the treatment of burn wounds. In 2016, Rashaan et al. published the results of their prospective study in which children with partial-thickness burns were treated. This study showed the potential benefits of alloplastic epidermal substitute treatment in terms of pain and scar formation. However, the study group consisted of only 21 patients [18]. In another study, the authors compared Suprathel^®^ to Mepilex^®^ Ag for the treatment of partial-thickness scalds in children. They reported no significant differences in any of the outcomes [33]. The results are consistent with our study’s, which showed that both the use of silver dressings and epidermal substitutes reduce the need for skin transplantation compared to classic paraffin gauze.

One of the most recent studies on this topic is by Schriek et al., in which they included a total of 2084 children treated for superficial and deep split-thickness skin burns. The study compared Suprathel^®^ with alternative dressings. The average number of procedures was statistically different, being 54.35% lower in the group of patients treated with Suprathel^®^ than in the control group (*p* < 0.0001). In the study group, 91.74% of children managed to be treated conservatively, compared to the control group, in which 23.95% of patients required split-thickness skin grafts [34]. In our study, the use of an epidermal substitute was associated with a greater number of interventions, but most of them concerned only a change in the upper layers of the dressing, not the substitute itself. When using classic dressings, each change involves intervening in the wound. The epidermal substitute remained on the wound until it was completely epithelialized. This allowed for a reduction in pain.

Our results demonstrated that using alloplastic epidermal substitution can shorten the hospital stay of children who have suffered severe burns. In our observation, the one-time application of synthetic epidermal substitutes lowered the necessity of skin grafting. Blome-Eberwein et al. reported their 4-year experience with the use of an epidermal substitute in superficial and deep second-degree burn wounds, including 138 pediatric patients. All of the wounds in this study healed without the need for skin grafting, and the average healing time was <14 days. As they pointed out, the use of an epidermal substitute allows for fewer dressing changes and easier overall management of the burn wound [35].

## 5. Conclusions

An epidermal substitute offers a good dressing option for burn wounds to improve the treatment and reduce the need for skin graft coverage. In the future, extended comparative or randomized trials are needed to confirm our results.

## Figures and Tables

**Figure 1 jcm-13-07238-f001:**
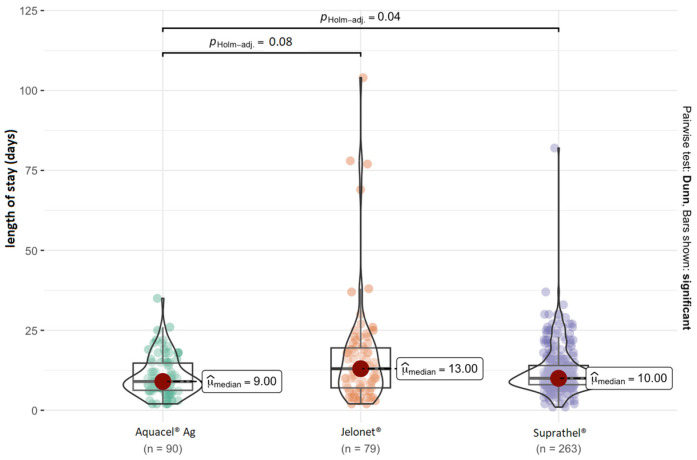
Length of hospitalization depending on the type of dressing (median, *p* values and density distribution).

**Figure 2 jcm-13-07238-f002:**
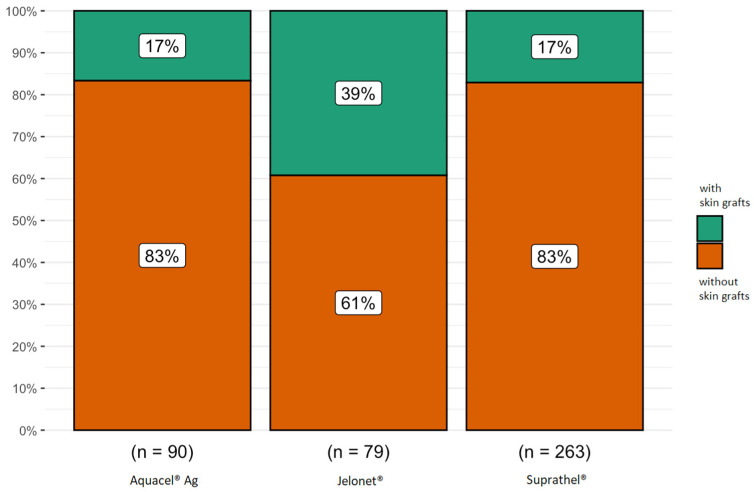
Skin grafts depending on the type of dressing (% of study group with and without skin transplantation).

**Table 1 jcm-13-07238-t001:** Characteristics of the study group.

Demographic characteristic	Number of patients	439
	Age (months, mean ± SD/range)	50 ± 56/6–216
	Sex (%)	M 63%/F 37%
Burn size and depth	TBSA Burn % (mean ± SD/range)	9 ± 9/0.5–79
	I/II(N/%)	58/13%
	II (N/%)	177/40%
	II/III (N/%)	190/43%
	III (N/%)	14/3%
Etiology	Scald (N/%)	339/77%
	Flame (N/%)	47/10.7%
	Electrical (N/%)	6/1.3%
	Burst (N/%)	18/4.1%
	Contact (N/%)	15/3.33%
	Oil (N/%)	15/3.33%
	Chemical (N/%)	1/0.24%
Intervention	Surgery (N/%)	415/95%
	Number of interventions (mean ± SD/range)	4.3 ± 3.2/1–29
	Skin graft (N/%)	92/22%
	Enzymatic debridement (N/%)	13/3%
	Fascial excision (N/%)	3/0.7%
	Alloplastic epidermal substitute (N/%)	263/63%
	Nano-silver dressing (N/%)	90/21%
	Paraffin gauze dressing (N/%)	80/19%

(N—number of patients; % percentage of the study group; LOS—length of hospital stay in days; TBSA—total body surface area).

## Data Availability

Data are contained within the article.

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
