# Peer review of "The Effectiveness of an Alloplastic Epidermal Substitute in the Treatment of Burn Wounds in Children: A Comparative Clinical Study of Skin Substitutes and Silver and Paraffin Gauze Dressings"

_jcm, 2024, doi:10.3390/jcm13237238_

Round 1

Reviewer 1 Report

Comments and Suggestions for Authors

The manuscript entitled 'Effectiveness of alloplastic epidermal substitute in the treatment of burn wounds in children: comparative clinical study of skin substitute, silver- and paraffin gauze dressing' reports comparative clinical study of a alloplastic epidermal substitute with silver- and paraffin gauze dressing in the treatment of burn wounds in children. 

These are few comments to revise:

1) In the introduction, can you please explain more about how burn wound and healing is different in children than from adults and how burn management is different from adults.  You can write on how paediatric skin structure, healing process, scarring, immune response, management is different than adults.

 2) In the materials and methods part, can you please add a study design section with clear subheadings like eligibility criteria with inclusion and exclusion criteria, sample size, intervention, wound management, study procedure, data collection, timelines, follow up, outcome analysis, quality control and data analysis plan. 

3) Discussion need to be more specific. Explain what factors would have contributed to the result.

4) Conclusion can be made into a single paragraph.

I wish the authors all the best.

Author Response

Dear Reviewer,

Thank you very much for all of your excellent remarks and for considering our paper for publication in the journal.

Those comments are valuable and very helpful for revising and improving our paper, as well as the important guiding significance to our researches. We have made correction which we hope will meet with approval.

Please find below my answers to your comments and enclosed the revised manuscript.

Yours sincerely,

Tomasz Korzeniowski

Comment 1: In the introduction, can you please explain more about how burn wound and healing is different in children than from adults and how burn management is different from adults.  You can write on how paediatric skin structure, healing process, scarring, immune response, management is different than adults.

Response 1: A paragraph on differences in burn wounds and healing between children and adults has been added in the introduction section as suggested.

Comment 2: In the materials and methods part, can you please add a study design section with clear subheadings like eligibility criteria with inclusion and exclusion criteria, sample size, intervention, wound management, study procedure, data collection, timelines, follow up, outcome analysis, quality control and data analysis plan.

Response 2: The materials and methods section has been modified to include study design and division into subheadings as recommended.

Comment 3: Discussion need to be more specific. Explain what factors would have contributed to the result.

Response 3: The discussion has been modified to be more specific. The results have been explained, related to other studies. Appropriate references have been added..

Comment 4: Conclusion can be made into a single paragraph.

Response 4: The conclusions section has been shortened to one paragraph as suggested.

Reviewer 2 Report

Comments and Suggestions for Authors

Dear authors,

I appreciate the chance to read and review your research as well as your contribution to wound healing society.

This study introduced valuable data regarding wound management in children. Introducing the correlation between age and burn degree in children can help other studies improve different aspects of burn wound prevention. Comparing different wound dressings with different healing properties while considering hospitalization can be an intriguing evaluation. This can assist scientists and physicians in selecting appropriate dressings based on the type of burn. I truly appreciate your methodology in analyzing and considering demographic factors, as well as introducing wound dressings and their functions in your study. The following comments can enhance the quality of your manuscript, making it more impactful and potentially publishable.

Major comments:

1-     One of the most important characteristics of burn wounds is the location of the burns. As you know, different areas in the body, such as the face with thinner skin or other parts of the body with thicker skin, have different healing patterns, so it would be better to include this criteria in your analysis.

2-     Regarding the clinical outcomes of your study, you need to consider several factors, including the infection rate, scarring, and the presence or absence of pain. Indeed, among over 400 children, some wounds became infected post-treatment, necessitating the monitoring of bacterial colonization beneath the dressings. Monitoring the impact of the dressing on keloid formation or hypertrophic scarring is crucial. Reducing the pain score is one of the most essential aspects of an appropriate dressing or wound treatment. As I read in the discussion section, you addressed these factors in some related studies, so I anticipate that the reader will search for similar data in your results.

3-     Another important factor in evaluating wound dressings is complications they may cause, such as irritation or allergic reactions. Moreover, in cases of 2nd and 3rd-degree burns, specific wound dressings or their renewal may cause burn contractures, primarily in the joints.

Minor Comments:

4-     Are the details in the introduction section, lines 63-66, relevant to the global context?

5-      In the Materials and Methods section, you mentioned the wound degrees. It may be better to explain wounds to readers unfamiliar with this classification.

6-     In Section 2.4 (statistical analysis), did you use any specific software for your evaluations, such as SPSS or Prism? If yes, please provide more details.

7-     In Table 1, you explained how you were intervening. It would be appropriate if you outlined the criteria used to select each intervention for patient treatment.

8-     In Figure 1, the age range is inconsistent. How can age represent a negative value? The labeling is not appropriate. For example, if someone simply glances at the graph, they might question how a person can age more than 130 years, despite the fact that the age is only 130 months. The labels on the x-axis are also inconsistent and might confuse the reader. (e.g., "Mosaic," "Superficial," "Second Degree," "Third Degree") It is difficult to distinguish different components, such as the median or interquartile range, in this figure. Perhaps by adding some labels, you could improve the clarity.

9-     The caption or legend in Figure 2 should contain the statistical data. Moreover, the p-values are in scientific notations. It would be more effective to write them in standard decimal form. The p-values might be better shown as asterisks (***) or by color-coding significant comparisons. We recommend providing a brief explanation of the violin plot evaluation process. Readers may struggle to understand the density distribution and different shapes.

10-   The caption or legend in Figure 3 should contain the statistical data. Moreover, the p-values are in scientific notations. It would be more effective to write them in standard decimal form. The labels 1 and 0 are difficult to understand. It's better to write with or without skin grafts instead.

Overall, this manuscript increased our understanding of wound management in children. Using the mentioned revisions, I believe it could impact burn wound management significantly. 

Author Response

Dear Reviewer,

Thank you very much for all of your excellent remarks and for considering our paper for publication in the journal.

Those comments are valuable and very helpful for revising and improving our paper, as well as the important guiding significance to our researches. We have made correction which we hope will meet with approval.

Please find below my answers to your comments and enclosed the revised manuscript.

Yours sincerely,

Tomasz Korzeniowski

Comment 1: One of the most important characteristics of burn wounds is the location of the burns. As you know, different areas in the body, such as the face with thinner skin or other parts of the body with thicker skin, have different healing patterns, so it would be better to include this criteria in your analysis.

Response 1: Location of the burns in the study group has been added in the results section as recommended.

Comment 2: Regarding the clinical outcomes of your study, you need to consider several factors, including the infection rate, scarring, and the presence or absence of pain. Indeed, among over 400 children, some wounds became infected post-treatment, necessitating the monitoring of bacterial colonization beneath the dressings. Monitoring the impact of the dressing on keloid formation or hypertrophic scarring is crucial. Reducing the pain score is one of the most essential aspects of an appropriate dressing or wound treatment. As I read in the discussion section, you addressed these factors in some related studies, so I anticipate that the reader will search for similar data in your results.

Response 2: Thank you for these valuable comments. The influence of the dressings used on the formation of hypertrophic scars and contractures is significant. The observation of the study group covered the period from admission to discharge from the hospital. We plan to conduct an analysis of long-term results in a prospective study. The epidermal substitute remained on the wound until epithelialization, and care was based only on replacing the top layers. We did not collect cultures and did not observe infection in this group of patients. I agree that adding a direct pain assessment would be valuable, which was not performed in this retrospective analysis. Pain reduction was related to the properties of epidermal substitute, which is presented in the discussion: “The epidermal substitute remained on the wound until it was completely epithelialized. This allowed for a reduction in pain.”

Comment 3: Another important factor in evaluating wound dressings is complications they may cause, such as irritation or allergic reactions. Moreover, in cases of 2nd and 3rd-degree burns, specific wound dressings or their renewal may cause burn contractures, primarily in the joints.

Response 3:  No irritation or allergic reactions were recorded in the study group. The results section has been updated with this data as suggested.

Comment 4: Are the details in the introduction section, lines 63-66, relevant to the global context?

Response 4:  The indicated details have been removed from the introduction section as suggested.

Comment 5: In the Materials and Methods section, you mentioned the wound degrees. It may be better to explain wounds to readers unfamiliar with this classification.

Response 5:  Information of burn depth assessment and classification has been added as recommended.

Comment 6: In Section 2.4 (statistical analysis), did you use any specific software for your evaluations, such as SPSS or Prism? If yes, please provide more details.

Response 6:  Information about the software used for statistics has been added as suggested.

Comment 7: In Table 1, you explained how you were intervening. It would be appropriate if you outlined the criteria used to select each intervention for patient treatment.

Response 7:  The type of intervention used in the study group was explained and updated in the wound management subsection.

Comment 8: In Figure 1, the age range is inconsistent. How can age represent a negative value? The labeling is not appropriate. For example, if someone simply glances at the graph, they might question how a person can age more than 130 years, despite the fact that the age is only 130 months. The labels on the x-axis are also inconsistent and might confuse the reader. (e.g., "Mosaic," "Superficial," "Second Degree," "Third Degree") It is difficult to distinguish different components, such as the median or interquartile range, in this figure. Perhaps by adding some labels, you could improve the clarity.

Response 8:  Due to inconsistency, the chart has been removed. The data contained in the chart has been presented in descriptive form.

Comment 9: The caption or legend in Figure 2 should contain the statistical data. Moreover, the p-values are in scientific notations. It would be more effective to write them in standard decimal form. The p-values might be better shown as asterisks (***) or by color-coding significant comparisons. We recommend providing a brief explanation of the violin plot evaluation process. Readers may struggle to understand the density distribution and different shapes.

Response 9: The figure has been modified as recommended.

Comment 10: The caption or legend in Figure 3 should contain the statistical data. Moreover, the p-values are in scientific notations. It would be more effective to write them in standard decimal form. The labels 1 and 0 are difficult to understand. It's better to write with or without skin grafts instead.

Response 10: The figure has been modified as recommended.

Round 2

Reviewer 2 Report

Comments and Suggestions for Authors

Dear authors,
Thank you for responding to my comments and amending the paper properly. I appreciate the effort you've made to improve the clarity and scientific rigor of your work. The edits improved the manuscript's quality and impact, and I am convinced it will make an addition to the field. I have no additional remarks.

Comments on the Quality of English Language

The quality of the English language in the manuscript is good and generally clear. However, I recommend carefully reviewing the sentence structures to ensure consistency and improve readability. A light proofreading for minor adjustments would further enhance the clarity of the text.